# *Drosophila* Temperature Preference Rhythms: An Innovative Model to Understand Body Temperature Rhythms

**DOI:** 10.3390/ijms20081988

**Published:** 2019-04-23

**Authors:** Tadahiro Goda, Fumika N. Hamada

**Affiliations:** 1Visual Systems Group, Abrahamson Pediatric Eye Institute, Division of Pediatric Ophthalmology, Cincinnati Children’s Hospital Medical Center, 3333 Burnet Avenue, Cincinnati, OH 45229, USA; 2Division of Developmental Biology, Cincinnati Children’s Hospital Medical Center, Cincinnati, OH 45229, USA; 3Department of Ophthalmology, College of Medicine, University of Cincinnati, Cincinnati, OH 45229, USA

**Keywords:** body temperature rhythms, Drosophila, DH31R, DH31, PDF, PDFR, circadian rhythms, locomotor activity rhythms, Calcr, Calcitonin receptor

## Abstract

Human body temperature increases during wakefulness and decreases during sleep. The body temperature rhythm (BTR) is a robust output of the circadian clock and is fundamental for maintaining homeostasis, such as generating metabolic energy and sleep, as well as entraining peripheral clocks in mammals. However, the mechanisms that regulate BTR are largely unknown. *Drosophila* are ectotherms, and their body temperatures are close to ambient temperature; therefore, flies select a preferred environmental temperature to set their body temperature. We identified a novel circadian output, the temperature preference rhythm (TPR), in which the preferred temperature in flies increases during the day and decreases at night. TPR, thereby, produces a daily BTR. We found that fly TPR shares many features with mammalian BTR. We demonstrated that diuretic hormone 31 receptor (DH31R) mediates *Drosophila* TPR and that the closest mouse homolog of DH31R, calcitonin receptor (Calcr), is essential for mice BTR. Importantly, both TPR and BTR are regulated in a distinct manner from locomotor activity rhythms, and neither DH31R nor Calcr regulates locomotor activity rhythms. Our findings suggest that DH31R/Calcr is an ancient and specific mediator of BTR. Thus, understanding fly TPR will provide fundamental insights into the molecular and neural mechanisms that control BTR in mammals.

## 1. Introduction

Most organisms exhibit approximately 24-h cycles of physiological and behavioral activity [1,2]. In humans, body temperature fluctuates by approximately 1 °C over the course of a day, increasing gradually during wakefulness and decreasing during sleep (Figure 1A) [3]. The daily fluctuation of body temperature (body temperature rhythm: BTR) is a robust output of the circadian clock [4,5,6]. BTRs are associated with other physiological functions, such as metabolism and sleep [7,8,9,10,11,12], and play an important role in regulating the entrainment of the peripheral clocks located in the lungs and liver [11,13]. However, the molecular and neural mechanisms by which the circadian clock regulates BTRs remain largely unclear. The fruit fly *Drosophila melanogaster* is a versatile model organism with powerful and sophisticated genetic toolkits and has been used to clarify complex biological systems, including circadian rhythms and sleep. In 2017, the Nobel Prize in Physiology or Medicine was given to Drs. Jeffrey C. Hall, Michael Rosbash, and Michael W. Young for their discoveries of mechanisms regulating circadian rhythms using *Drosophila*. We first asked whether *Drosophila* can be used as a model organism for BTRs. We found that flies exhibit robust temperature preference behavior and that their preferred temperature fluctuates over a 24-h period. Because flies are small ectotherms and their body temperatures are very close to the ambient temperatures in the environments they choose, this fluctuation in preferred temperature can give rise to a BTR in flies [7,14,15,16,17,18]. The evidences in humans and rats suggest that BTRs are controlled separately from locomotor activity rhythms [7,19,20,21,22]. We found in *Drosophila* that BTR is controlled separately from the locomotor activity rhythms as well [18]. Furthermore, we recently showed that a G-protein-coupled receptor (GPCR), the calcitonin receptor, modulates BTRs but not locomotor activity rhythms in both mammals and flies [23]. Thus, our data suggest that there are evolutionarily conserved mechanisms that contribute to the circadian regulation of body temperature in mammals and flies. Here, we describe our current knowledge of the regulatory mechanisms underlying BTRs in *Drosophila* and mice and discuss how *Drosophila* can be a model organism for BTRs.

## 2. *Drosophila* Exhibit a Robust Temperature Preference Rhythm (TPR)

To determine the molecular and neural mechanisms by which the circadian clock regulates BTRs, we first designed a behavioral experiment to monitor the BTR in *Drosophila*. Because *Drosophila* are small ectotherms, their body temperatures are very close to the temperature of their surroundings [7,14,15,16,17]. Therefore, we sought to monitor their surrounding temperatures and conduct a behavioral assay of temperature preference (Figure 2) [24,25,26]. *Drosophila* exhibit robust temperature preference behavior in 12:12-h light-dark (LD) cycles; they avoid warm and cold temperatures and prefer ~25 °C in the morning, which suggests that the flies’ body temperatures are very close to ~25 °C at that time [25]. We monitored their preferred temperatures for 24 h and found that the preferred temperatures fluctuated over that period; the preferred temperature was 25 °C in the morning, then increased during the daytime and decreased during the night (Figure 1B) [18]. We refer to this rhythmic change in preferred temperature as temperature preference rhythm (TPR). Given that the body temperature in *Drosophila* is close to their ambient temperature, the TPR results in a *Drosophila* BTR. Importantly, the *Drosophila* TPR shows a rhythmic pattern similar to that of the human BTR [3] (Figure 1). Furthermore, we demonstrated that TPR is a clock-controlled behavior based on the following observations: we showed that the fluctuation of preferred temperature was still maintained in constant darkness [27] and that two clock mutants—*timeless* and *period* mutant flies (*tim*^01^ and *per*^01^, respectively)—lost TPR in both LD and DD [18]. The data suggest that TPR is not solely generated by a response to light inputs but is regulated by an endogenous circadian clock. Thus, we discovered that *Drosophila* exhibit a robust TPR, which generates a BTR in flies.

## 3. TPR is Controlled by the Noncanonical Clock Circuits via Dorsal Neurons 2 (DN2s)

Because several researches in humans and rats suggest that BTR is controlled separately from locomotor activity rhythms [7,19,20,21,22], we asked whether *Drosophila* TPR is also controlled separately from locomotor activity rhythms. There are ~150 central pacemaker neurons in the fly brain, and they have functions similar to those of mammalian suprachiasmatic nucleus (SCN) neurons [28]. Based on their locations in the brain, pacemaker neurons are divided into four groups of lateral neurons (small ventrolateral neurons (sLNvs), large LNvs (lLNvs), dorsolateral neurons (LNds), and lateral posterior neurons (LPNs)) and three groups of dorsal neurons (DN1s, DN2s, and DN3s) [29]. We demonstrated that DN2s are the main clock neurons that regulate TPR, but DN2s do not regulate locomotor activity rhythms [18]. Thus, TPRs and locomotor activity rhythms are regulated by distinct pathways. Interestingly, the TPR persists when constant light (LL) is maintained for eight days, whereas locomotor activity rhythms are disrupted by four days of LL conditions [18]. This result also implies that TPR and locomotor activity rhythms are separately regulated.

## 4. Molecular and Neural Mechanisms Regulating TPR

To understand the molecular mechanisms of TPR, we focused on neuropeptides and receptors because they control many behaviors and physiological activities. In this review, we summarize the clock neurons and neuropeptide signaling that control the TPR at different times of the day: daytime (zeitgeber time (ZT) 1–12), night onset (ZT 10–15) and before dawn (ZT 22–24) (Figure 3).

### 4.1. Diuretic Hormone 31 Receptor (DH31R) Regulates the Daytime TPR (ZT 1-12)

The neuropeptide pigment-dispersing factor (PDF) and its receptor (PDFR) play important roles in the synchronization of molecular rhythms among core clock neurons and maintaining robust locomotor activity rhythms in *Drosophila* [30,31,32,33,34,35]. Therefore, we initially anticipated the involvement of PDF-PDFR signaling in TPR. However, both *Pdf* mutant (*Pdf*^01^) and *Pdfr* mutant (*Pdfr*^5304^) flies exhibited a normal increase in preferred temperatures during the daytime [23], suggesting that PDF and PDFR do not have a main role in regulating the daytime TPR.

DH31R belongs to the class B G-protein coupled receptor (GPCR) family, the same family as PDFR [36]. The ligand DH31 can activate DH31R as well as PDFR [34,37,38], and DH31 is also involved in sleep before dawn [39]. Therefore, we next asked whether DH31R plays an important role in the TPR. We found that *Dh31r* mutant (*Dh31r*^1^^/*Df*^) flies exhibited a constant preferred temperature of approximately 27 °C during the daytime (Figure 4) [23]. The *Dh31r* mutant also disrupted the daytime TPR in DD, suggesting that the function of DH31R in the daytime TPR is associated with the endogenous circadian clock.

Because DN2s are the main clock neurons for the TPR [18], we expected that DH31R might be expressed in DN2s. However, immunostaining with an anti-DH31R antibody revealed that DH31R is not expressed in DN2s but is expressed in a subset of DN1s, DN3s, and lLNvs [23]. To determine whether DH31R expression in these clock neurons is important for the TPR, we performed rescue experiments and found that *Dh31r* expression in all clock neurons restored the TPR. This result suggests that DH31R in clock cells is sufficient for regulating the daytime TPR (Figure 7A).

How does DH31R in clock cells regulate the daytime TPR? We recognized two possibilities, as follows: (1) DH31R controls the molecular oscillations of clock genes; or (2) DH31R acts on the downstream output of clock neurons. To examine these possibilities, we monitored the molecular oscillations in core clock neurons, including DN2s, and found that they were intact in *Dh31r* mutant flies. These data suggest that DH31R regulates the daytime TPR by acting downstream of clocks [23]. Notably, we also demonstrated that *Dh31r* mutant flies exhibited normal locomotor activity rhythms [23]. These data provide further support for the idea that TPR and locomotor activity rhythms are regulated separately.

### 4.2. Both DH31 and PDF Contribute to the Regulation of the Daytime TPR (ZT 1-12)

Next, we asked whether mutants of *Dh31*, a ligand of DH31R, showed an abnormal daytime TPR. However, *Dh31*, as well as *Pdf* mutants, showed a normal daytime TPR [40]. Given that both *Dh31* and *Pdf* single mutants showed a normal daytime TPR, we hypothesized that DH31 and PDF might have a redundant function. To examine this possibility, we created *Dh31* and *Pdf* double mutant flies and found that they had a disrupted daytime TPR, with a constant preferred temperature during the daytime [23]. To determine whether DH31 and PDF act on the clock neurons, we performed rescue experiments using membrane-tethered peptides, which localize on the cell surface and prevent spreading [41]. We demonstrated that abnormal TPR in *Dh31* and *Pdf* double mutant flies were recovered by the expression of either *tethered-Dh31* (*t-Dh31*) or *tethered-Pdf* (*t-Pdf*) in all clock cells [23]. These data suggest that both DH31 and PDF act on the clock cells to regulate the daytime TPR. It would be of interest to further investigate the relationships between DH31, PDF, and DH31R in terms of the regulation of the TPR.

### 4.3. DH31 Acts on DN2s Through PDFR, but not DH31R, to Regulate the Night-onset TPR (ZT 10-15)

The preferred temperature for *Drosophila* drops dramatically by approximately 1.5 °C at the transition from day to night (night-onset TPR) (Figure 3). In *Dh31* mutant flies, the dramatic decrease in preferred temperature at the transition from day to night was dampened to 0.6 ~ 0.8 °C (Figure 5D) [40], suggesting that DH31 is responsible for the night-onset TPR. To identify the neurons that regulate the night-onset TPR through DH31, we expressed t-DH31 in a subset of clock cells in *Dh31* mutant flies. We demonstrated that t-DH31 expression in DN2s rescued the decrease in preferred temperature at the transition from day to night in *Dh31* mutant flies, suggesting that DH31 signaling acts on DN2s to regulate the night-onset TPR.

What is the receptor of DH31 for the regulation of the night-onset TPR? We found that *Pdfr* mutant flies exhibited a dampened night-onset TPR (Figure 5B) [40]. Because it has been shown that DH31 can activate PDFR [34,37,38], we speculated that DH31 activates PDFR to regulate the night-onset TPR. To test this possibility, we expressed PDFR in a subset of clock cells, including DN2s, in *Pdfr* mutant flies. We found that PDFR expression in DN2s restored a robust night-onset TPR, indicating that PDFR in DN2s is sufficient for the night-onset TPR. Taken together, these data suggest that DH31 acts on DN2s through PDFR to regulate the night-onset TPR (Figure 7B). Importantly, although DH31 activates DH31R, the *Dh31r* mutants exhibit a normal night-onset TPR. Because DH31R is required for the daytime TPR but not for the night-onset TPR, it appears that the daytime and night-onset TPRs are regulated by different mechanisms.

### 4.4. PDF Neurons Regulate the Predawn TPR via DN2s (ZT 22-24)

In the wild type flies, the preferred temperature before dawn (ZT22–24) is similar to that of early morning at ZT1-3 (predawn TPR). To understand the mechanisms of the predawn TPR, we first focused on the PDF neurons because they regulate morning anticipation in locomotor activity rhythms; *Pdf* mutant flies show a lack of morning anticipation, a phenomenon that is manifested as an increase in locomotor activity before dawn [42]. Therefore, we asked whether PDF is involved in regulating TPR before dawn. We found that *Pdf* mutant flies preferred a lower nighttime temperature (ZT 16–24) than wild-type flies (Figure 6) [43]. A similar phenotype was observed in the flies whose PDF neurons were inhibited by expressing the mammalian inward rectifier K^+^ channel *Kir2.1* [44]. These data suggest that PDF neurons play an important role in regulating the predawn TPR.

Because PDF neurons are not the main pacemaker neurons for the TPR [18], we wondered what roles PDF neurons might play in regulating the predawn TPR. Because sLNvs, a subset of PDF neurons, are known to contact DN2s in larvae [45] and project near DN2s in adult, the main pacemaker neurons for the TPR [46], we hypothesized that sLNvs might contact DN2s to regulate the predawn TPR. Using green fluorescent protein (GFP) reconstitution across synaptic partners (GRASP) analysis [47,48,49], we showed that sLNvs make contacts with DN2s and that the number of contacts fluctuates over 24 h, with a peak before dawn [43].

Using a genetically encoded calcium indicator, GCaMP with the adenosine triphosphate (ATP)-gated ionotropic purinoreceptor, P2X2 [50], we showed that sLNvs functionally contact and activate DN2s [43]. The data suggest that sLNvs and DN2s have a temporal interaction and that sLNvs manipulate the activity of DN2s. However, the role of DN2s other than as the main clock remained unclear. Therefore, we examined how DN2 activity affects the TPR. We tested the TPR in flies whose DN2s were inhibited by *Kir2.1* [44]. Our data showed that the inhibition of DN2s caused a reduced preferred temperature at all times in the circadian cycle compared to that of the control flies [43]. Thus, it is likely that sLNvs activate DN2s to increase the preferred temperature at dawn. Together, our data suggest that the temporal interaction between sLNvs and DN2s peaks before dawn and that sLNv-DN2 neural circuits regulate the setpoint of preferred temperature before dawn (Figure 7C).

### 4.5. Anterior Cell (AC) Neurons Regulate the Predawn TPR via sLNvs (ZT 22-24)

How are the clock neurons interacted with the temperature-sensing neurons? One of the *Drosophila* transient receptor potential A (TRPA) superfamilies, TRPA1, which is activated above 25 °C, is expressed in anterior cell (AC) neurons located in the brain along with the antennal nerve [25,51]. TRPA1 in AC neurons is responsible for temperature preference behavior [25].

Because both AC neurons and sLNvs project to the dorsal protocerebrum [25,46,52,53], we asked whether sLNvs interact with AC neurons. Using GRASP, we demonstrated that sLNvs have contacts with AC neurons [43]. To determine whether sLNv-AC circuits are the important neural circuits for the predawn TPR, we asked whether AC inhibition causes a lower preferred temperature phenotype similar to that of sLNv-inhibited flies. We demonstrated that either *TrpA1* knockdown in AC neurons or inhibition of AC neurons by *Kir2.1* resulted in a lower preferred temperature from late night until just before dawn (ZT 19–24), which is a phenotype similar to the one resulting from *Pdf* mutation or inhibition of PDF neurons. The data suggest that AC neurons regulate the temperature setpoint before dawn and that sLNv-AC circuits are the important neural circuits for the predawn TPR. Given that AC neurons are serotonergic [52] and LNvs express one of the serotonin receptors, 5-hydroxytryptamine receptor 1B, 5HT1B [54], we asked whether serotoninergic transmission from AC neurons to sLNvs contributes to the regulation of the predawn TPR. We found that knocking down 5HT1B in sLNvs caused a reduced preferred temperature before dawn, which is similar to the phenotype observed in AC- or sLNv-inhibited flies. Therefore, our data suggest that the sLNv-AC neural circuits regulate the predawn TPR via serotonin signaling.

## 5. Mammalian BTRs

Human body temperatures increase gradually during wakefulness and decrease during sleep (Figure 1A) [3,4,5,6]. BTR is controlled by the circadian clock, balancing heat production and heat loss for 24 h via physiological and behavioral approaches [55,56]. As daily variations in BTR are robust and parallel fluctuations in locomotor activity rhythms, BTR is widely used to monitor circadian rhythms in mammals. The molecular mechanisms that regulate BTR remain largely uncharacterized, although a study in which subsets of neurons in rats were surgically ablated suggested that locomotor activity rhythms and BTRs are controlled by different output pathways that originate from the suprachiasmatic nucleus (SCN) [19]. In humans, body temperature fluctuates even when locomotor activity is restricted [20,21], and BTR and locomotor activity rhythms can be experimentally dissociated, a phenomenon known as spontaneous internal desynchronization [22,57]. These data suggest that the BTR is controlled in a manner distinct from that of locomotor activity rhythms. However, molecular evidence supporting this possibility has not been reported. Therefore, there was—and remains—a critical need to identify mechanisms that regulate BTRs.

## 6. The Calcitonin Receptor Regulates the BTR in Mice

Given that many fundamental biological mechanisms are functionally conserved between flies and mammals, we asked whether a mammalian homolog of DH31R was also involved in the regulation of mammalian BTRs. The closest mammalian homologs of *Drosophila* DH31R are the calcitonin receptor (Calcr) and the calcitonin receptor-like receptor (Calcrl) [36]. Both Calcr and Calcrl are seven-transmembrane-domain class B GPCRs and are involved in physiological functions including calcium metabolism [58,59,60]. The amino acid sequence similarity between DH31R and Calcr or Calcrl is 67.9% or 67.4%, respectively, in the seven transmembrane domains and 74.6% or 81.0%, respectively, in the N-terminal region [23]. Because Calcr is expressed in the SCN in rats and mice, whereas Calcrl is not [61,62,63], we focused on Calcr and examined its role in the BTR using mice.

To determine where Calcr is expressed in the SCN, we performed in situ hybridization as well as immunohistochemistry using an anti-Calcr antibody. We demonstrated that *Calcr* mRNA and Calcr protein were similarly distributed in the dorsomedial area of the SCN, a region corresponding to the SCN shell. Arginine vasopressin (AVP) and vasoactive intestinal polypeptide (VIP) are markers of the SCN shell and SCN core, respectively [64]. We performed double immunostaining with anti-Calcr and anti-AVP or anti-VIP antibodies and found that AVP and Calcr partially overlapped in the SCN shell, whereas VIP and Calcr did not colocalize. These data also suggest that Calcr is expressed in the SCN shell but not in the VIPergic SCN [23].

To examine whether Calcr plays an important role in the circadian rhythm of body temperature, we compared the BTRs of wild-type and *Calcr* knockout (KO) mice [23]. The body temperature of wild-type mice fluctuates over a 24-h period [3]. Because mice are nocturnal animals, their BTR patterns are different from those of humans. During the daytime, when mice are primarily resting, their body temperature gradually decreases during the early phase and increases during the late phase [65,66,67]. However, during the night (the active phase of mice), the animals’ body temperature displays bimodal peaks in the early night and at dawn, with a deep trough late at night (i.e., the midnight trough) [67,68]. Although we found that the body temperatures of both wild-type and Calcr KO mice fluctuated over a 24-h period, they were significantly different at midnight. Specifically, the body temperatures of wild-type mice showed a deep trough at midnight, whereas the body temperatures of Calcr KO mice lost the characteristic trough and remained relatively unchanged during the night (Figure 8A). These data indicate that the lack of Calcr expression causes a shallow midnight trough in body temperature, suggesting that Calcr is required for body temperature fluctuations, particularly during the night (the active phase of mice). Importantly, this finding is consistent with the findings for the TPR phenotype of the *Drosophila Dh31r* mutant, as we observed that the *Dh31r* mutation caused a flat TPR during the daytime (the active phase of flies). Therefore, both *Drosophila* DH31R and mouse Calcr are required for body temperature fluctuation during the active phase [23].

Because body temperature may be increased by locomotor activity, it is possible that increased levels of locomotor activity result in the elevated body temperatures observed in *Calcr* KO mice. To examine this possibility, we compared fluctuations in locomotor activity between wild-type and *Calcr* KO mice; however, there was no significant difference in locomotor activity between the two groups (Figure 8B). Therefore, we concluded that Calcr specifically mediates body temperature fluctuations during the night and does not affect locomotor activity rhythms.

Together, our data demonstrated that DH31R mediates the *Drosophila* TPR and that calcitonin Calcr is essential for the normal BTR in mice [23]. Importantly, neither DH31R nor Calcr regulates locomotor activity rhythms. Although the mechanisms that underlie thermoregulation in *Drosophila* and mammals are completely different, our data identify the calcitonin receptors DH31R and Calcr as fundamental, ancient mediators of daily BTRs in both flies and mice. Thus, the *Drosophila* TPR is a functionally conserved model for mammalian BTRs, and our study using the *Drosophila* TPR will provide fundamental insights into the molecular and neural mechanisms that control BTRs in mammals.

## 7. Conclusions

BTRs are essential for homeostatic functions and are regulated separately from locomotor activity rhythms [7]. Our studies over the past several years established that the molecular and neural mechanisms that control the fly TPR resemble those of the mammalian BTR [18,26,40,43]. We also determined that calcitonin receptors (fly DH31R and mouse Calcr) are ancient regulators of the fly TPR and the mouse BTR [23]. The powerful genetic tools available in flies will allow us to reveal mechanisms that have remained a long-standing mystery: the mechanisms that control how body temperature increases during wakefulness and decreases during sleep. Thus, our expectation is that the fly TPR will serve as an innovative model to understand the mammalian BTR. The outcome of this research is expected to establish important foundations for our understanding of the daily rhythms of body temperature in both flies and mammals, bearing important implications for the treatment of circadian clock diseases, sleep problems, metabolic diseases, and the health of night-shift workers.

## Figures and Tables

**Figure 1 ijms-20-01988-f001:**
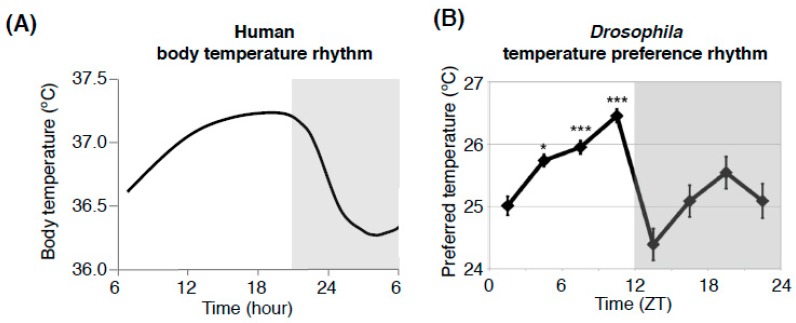
Comparison between the human body temperature rhythm (BTR) and the *Drosophila* temperature preference rhythm (TPR). The human BTR (**A**) (replotted the figure using an example form Duffy JF et al. 1998, Figure 3 [3]) and the *Drosophila* TPR (**B**) (modified from Kaneko et al. 2012, Figure 1 [18]). The shadow area in (**A**) shows the sleep period, and (**B**) shows the dark period. ZT: zeitgeber time. *: *p* < 0.05, ***: *p* < 0.001 (comparison to ZT 1–3). Given that the body temperature of *Drosophila* is close to their ambient temperature, the TPR results in fluctuations in body temperature. The TPR shows a rhythmic pattern similar to that of the human BTR.

**Figure 2 ijms-20-01988-f002:**
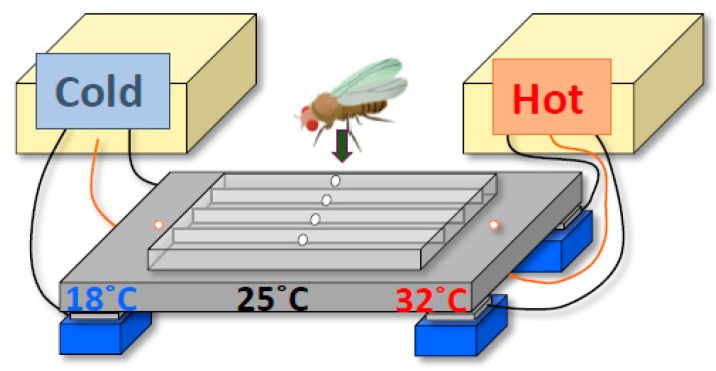
*Drosophila* temperature preference behavioral assay. A temperature gradient from 18 to 32 °C is generated in a chamber made with a metal plate and a plexiglass cover. Flies are introduced into the chamber through the holes in the cover. Within 30 minutes, the flies settle in the locations with their preferred temperatures. Because their body temperatures are very close to the ambient temperature, their body temperatures can be determined by measuring the temperature in the place where they are located. The diagram is modified from Goda et al. 2014, Figure 3 [26].

**Figure 3 ijms-20-01988-f003:**
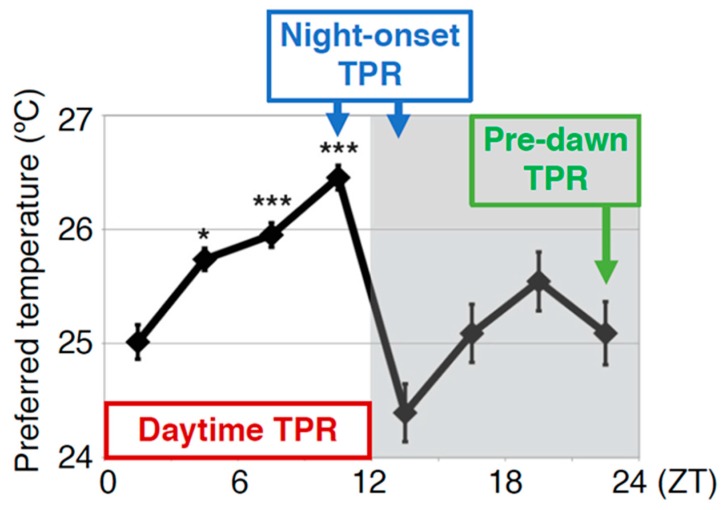
The TPR is regulated by different regulatory mechanisms at different times of the day: daytime, night onset, and predawn. Daytime TPR: The increase in preferred temperature during the daytime (ZT 1–12: shown by the red rectangle). Night-onset TPR: The dramatic decrease in preferred temperature at the transition from day to night (ZT 10–15: shown by blue rectangle). Predawn TPR: The preferred temperature just before dawn is similar to that of early morning (ZT 22–24: shown by green rectangle). The TPR is regulated by different regulatory mechanisms in each of these periods. The graph is modified from Kaneko et al. 2012, Figure 1 [18]. *: *p* < 0.05, ***: *p* < 0.001 (comparison to ZT 1–3).

**Figure 4 ijms-20-01988-f004:**
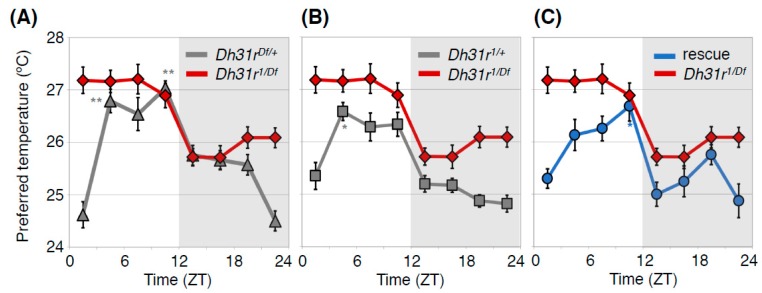
DH31R regulates the daytime TPR. Comparison of TPR between *Dh31r* mutant flies (*Dh31r*^1^^/*Df*^, red line in (**A**–**C**) and heterozygous control flies (*Dh31r*^1/+^, gray line in (**A**) or *Dh31r^Df/+^*, gray line in (**B**)) or *Dh31r* genomic rescue flies (rescue, blue line in (**C**)). *Dh31r* mutant flies show an abnormal daytime TPR, in which they prefer a constant temperature of approximately 27°C during the daytime. The control flies exhibit a normal daytime TPR, in which their preferred temperature increases during the daytime (**A**,**B**). Genomic rescue of *Dh31r* flies restored normal daytime TPR (**C**). *Dh31r*^1^ is a P-element insertion mutant (*PBac {WH}Dh31-R^f05546^*), and *Dh31r^Df^* is a deletion mutant [*Df(2R) BSC273*]. In the *Dh31r* genomic rescue fly, the Ch321-57F06 BAC clone, which includes the entire *Dh31r* gene region, is inserted into the genome. The graphs are modified from Goda et al. 2018, Figure 1 [23]. *: *p* < 0.05, **: *p* < 0.01 (comparison to ZT 1–3).

**Figure 5 ijms-20-01988-f005:**
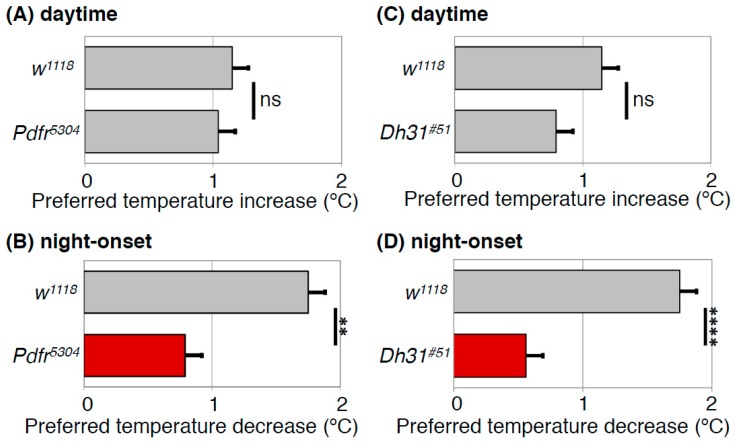
PDFR and DH31 regulate the night-onset TPR. Comparison of daytime TPR (**A**,**C**) or night-onset TPR (**B**,**D**) between *w*^1118^ and *Pdfr* mutant (*Pdfr*^5304^) (**A**,**B**) or *Dh31* mutant (*Dh31*^#51^) (**C**,**D**) flies. Both *Pdfr*^5304^ and *Dh31*^#51^ flies show significantly dampened night-onset TPR (**B**,**D**), while both exhibit robust increases in preferred temperature during the daytime (**A**,**C**). The data suggest that both PDFR and DH31 are responsible for the night-onset TPR. The graphs are modified from Goda et al. 2016, Figure 1 and Figure 3 [40]. t-test: **: *p* < 0.01, ****: *p* < 0.0001.

**Figure 6 ijms-20-01988-f006:**
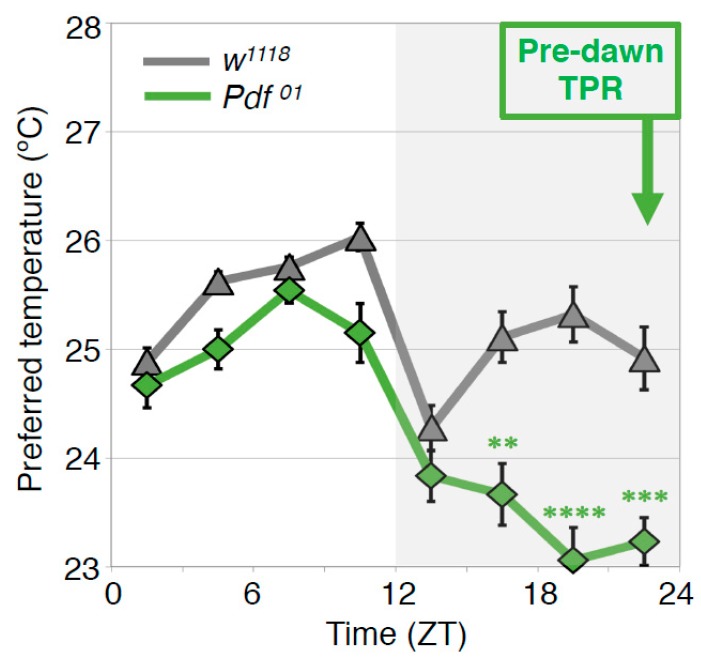
PDF regulates the predawn TPR. Comparison of TPRs between *w*^1118^ and *Pdf* mutant (*Pdf*^01^) flies. *Pdf*^01^ flies exhibit robust daytime (ZT 1–12) and night-onset TPRs (ZT 10–15). However, their preferred temperatures continue decreasing during the night and are much lower before dawn (ZT 22–24) than that of *w*^1118^ flies (predawn TPR). These data suggest that PDF is important for the predawn TPR. t-test between *w*^1118^ and *Pdf*^01^ in each ZT: **: *p* < 0.01, ***: *p* < 0.001, ****: *p* < 0.0001.

**Figure 7 ijms-20-01988-f007:**
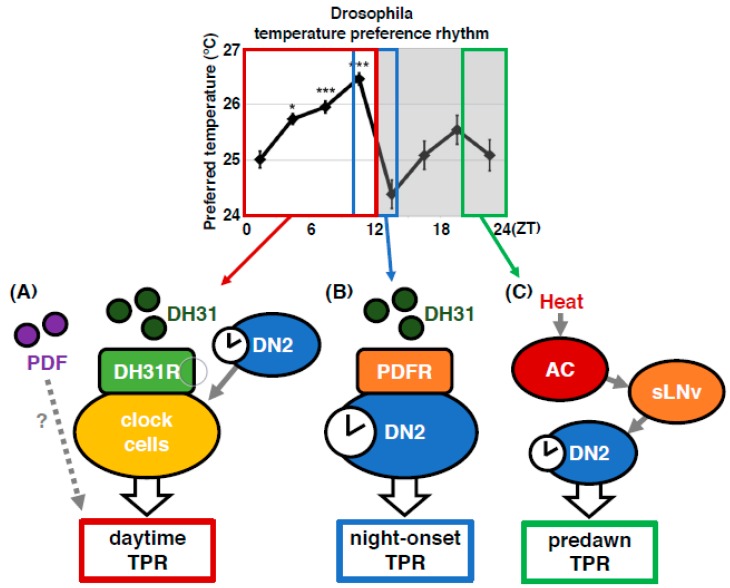
The *Drosophila* TPR is regulated by different mechanisms at different times of the day. Models for regulatory mechanisms of the TPR in each time range: daytime (ZT 1–12), night onset (ZT 10–15), and predawn (ZT 22–24). The daytime TPR: DN2s are the main clock neurons. DH31 acts on DH31R in a subset of clock neurons to regulate the daytime TPR. PDF is also involved in daytime TPR regulation. The night-onset TPR: DH31 acts on DN2s via PDFR to regulate the night-onset TPR. The predawn TPR: An AC-sLN-DN2 neural circuit regulates the proper setting of temperature preference before dawn.

**Figure 8 ijms-20-01988-f008:**
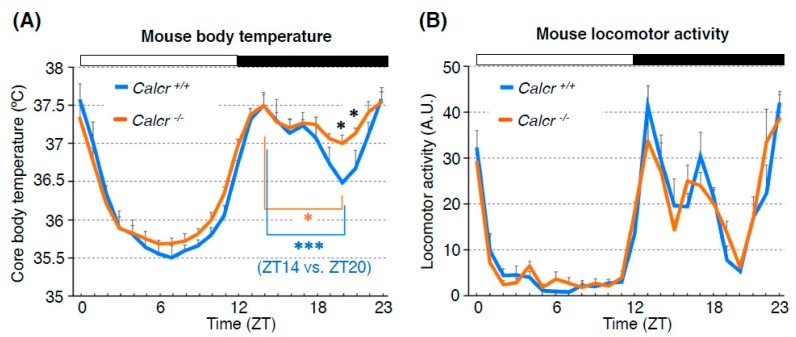
Calcr contributes to body temperature regulation during the night (the active phase for mice). Mouse BTR (**A**) and locomotor activity rhythm (**B**) in control (*Calcr*^+/+^: blue line) and *Calcr* knockout (*Calcr*^−/−^: orange line) mice. The body temperatures of wild-type mice showed a deep trough at midnight, whereas the body temperatures of *Calcr* knockout mice lost the characteristic trough and remained relatively unchanged during the night (**A**). The locomotor activity rhythm in *Calcr* knockout did not show a significant difference from that of the control, suggesting that *Calcr* is responsible for mammalian body temperature regulation but not locomotor activity rhythm during the night (the active phase for mice). White and black bars on the graphs indicate the 12-h light and dark phases. The graphs are modified from Goda et al. 2018, Figure 6 [23]. *: *p* < 0.05, ***: *p* < 0.001.

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
