# Peer review of "Drosophila Temperature Preference Rhythms: An Innovative Model to Understand Body Temperature Rhythms"

_ijms, 2019, doi:10.3390/ijms20081988_

Reviewer 1 Report

This is a nice specialized review on temperature preference rhythms in drosophila that also describes how it is linked to body temperature rhythms in mammals. The work in the drosophila TPR has been done essentially by the group of the authors and they nicely describe their work.

I just have two small comments:

P 5 (bottom) says that PDF and DH31 act redundantly to control daytime TPR with DH31R (Goda 2016). However, figure 7 only indicates DH31.

P 9: internal desynchronization between body temperature and activity refers to a 2001 review.  More recent and specific work should be cited (example: cambras et al PNAS 2007)

Author Response

We thank the reviewers for their critiques; they have been very helpful in substantially improving the manuscript. The reviewers’ comments are in italics, and our responses are in bold typeface.

Reviewer 1 comments:

This is a nice specialized review on temperature preference rhythms in Drosophila that also describes how it is linked to body temperature rhythms in mammals. The work in the Drosophila TPR has been done essentially by the group of the authors and they nicely describe their work.

I just have two small comments:

P 5 (bottom) says that PDF and DH31 act redundantly to control daytime TPR with DH31R (Goda 2016). However, figure 7 only indicates DH31.

We revised Fig. 7A. We also added the following sentence in the figure legend of Fig. 7A; “PDF is also involved in daytime TPR regulation.”

 P 9: internal desynchronization between body temperature and activity refers to a 2001 review.  More recent and specific work should be cited (example: cambras et al PNAS 2007)

We added a citation of the paper of Cambras et al PNAS 2007 on page 9.

Reviewer 2 Report

In this manuscript, Goda and Hamada present a very detailed review of the findings that the group has obtained in recent years in relation to the knowledge of temperature preference rhythms (TPR) in Drosophila and the molecular and neural mechanisms by which the circadian clock regulates them. These mechanisms resemble those of the body temperature rhythm (BTR) of mammals, which are additionally matter of this review.

The authors focus the Drosophila TPR as an output of the circadian clock that is associated with the body temperature rhythm (BTR) since this is a small ectotherm and its body temperature is very close to the environmental temperature that the animal chooses. The rhythms of TPR are essential for the maintenance of body homeostasis and are regulated by the circadian clock of the dorsal neurons DN2s of the brain's main clock in Drosophila. This regulation occurs differentially to the rhythms of locomotor activity. Additionally, the authors highlight the determinant role that the diuretic hormone receptor (DH31R) has in regulating the TPR of Drosophila, with different mechanisms acting in three daily phases: daytime period, the night onset, and the pre-dawn. The molecular and neural mechanisms that control fly TPR also resemble those of the mammalian BTR, in which the closest homologous of Drosophila DH31R are the calcitonin receptors (Calcr), which is expressed in the central SCN clock in rats and mice. Studies in Calcr KO mice demonstrate that Calcr is required for body temperature fluctuation, being essential for normal BTR in mice. All this shows the calcitonin receptors (DH31H, Calcr) as ancient and specific regulators of TPR (fly) and BTR (mouse). The authors conclude that fly TPR can serve as an innovative model to understand the BTR of mammals and that the powerful genetic tools available in flies will accelerate the knowledge of these mechanisms also in mammals.

In general, the manuscript is well written and organized, and is easy to read and understand, even for someone who is not a specialist in the subject in question. From this point of view, the work constitutes a valuable document that brings together advances in the knowledge, at the molecular and neural level of the regulatory mechanisms of the body temperature rhythm, presenting the Drosophila BTR as a valuable model to obtain information that can be applicable to the BTR in mammals. In this sense I only include a few comments that may help to improve the manuscript.

In some sections, several citations are included in the beginning of each paragraph in relation to the specific topic. Subsequently, remarked results of the research carried out on the subject are described, but this is not accompanied of specific citations that could facilitate the follow-up of these investigations. For instance, in point 4.5 regarding the implication of serotonergic transmission in the regulation of predawn TPR it would be appropriate to include citations of individual works that endorse the results. Something similar occurs in section 6, which refers to the involvement of the calcitonin receptor in the regulation of BTR in mice. At the end of paragraph 3 of this section it would be necessary to specify to which previous works the results cited in the text correspond.

Figure 2 should be located later in the text, since part of its content, for example, the mechanisms that regulate the pre-dawn are exposed later in the text. A suitable place would be just before section 5. Mammalian BTR.

Author Response

We thank the reviewers for their critiques; they have been very helpful in substantially improving the manuscript. The reviewers’ comments are in italics, and our responses are in bold typeface.

Reviewer 2 comments:

In this manuscript, Goda and Hamada present a very detailed review of the findings that the group has obtained in recent years in relation to the knowledge of temperature preference rhythms (TPR) in Drosophila and the molecular and neural mechanisms by which the circadian clock regulates them. These mechanisms resemble those of the body temperature rhythm (BTR) of mammals, which are additionally matter of this review.

The authors focus the Drosophila TPR as an output of the circadian clock that is associated with the body temperature rhythm (BTR) since this is a small ectotherm and its body temperature is very close to the environmental temperature that the animal chooses. The rhythms of TPR are essential for the maintenance of body homeostasis and are regulated by the circadian clock of the dorsal neurons DN2s of the brain's main clock in Drosophila. This regulation occurs differentially to the rhythms of locomotor activity. Additionally, the authors highlight the determinant role that the diuretic hormone receptor (DH31R) has in regulating the TPR of Drosophila, with different mechanisms acting in three daily phases: daytime period, the night onset, and the pre-dawn. The molecular and neural mechanisms that control fly TPR also resemble those of the mammalian BTR, in which the closest homologous of Drosophila DH31R are the calcitonin receptors (Calcr), which is expressed in the central SCN clock in rats and mice. Studies in Calcr KO mice demonstrate that Calcr is required for body temperature fluctuation, being essential for normal BTR in mice. All this shows the calcitonin receptors (DH31H, Calcr) as ancient and specific regulators of TPR (fly) and BTR (mouse). The authors conclude that fly TPR can serve as an innovative model to understand the BTR of mammals and that the powerful genetic tools available in flies will accelerate the knowledge of these mechanisms also in mammals.

In general, the manuscript is well written and organized, and is easy to read and understand, even for someone who is not a specialist in the subject in question. From this point of view, the work constitutes a valuable document that brings together advances in the knowledge, at the molecular and neural level of the regulatory mechanisms of the body temperature rhythm, presenting the Drosophila BTR as a valuable model to obtain information that can be applicable to the BTR in mammals. In this sense I only include a few comments that may help to improve the manuscript.

In some sections, several citations are included in the beginning of each paragraph in relation to the specific topic. Subsequently, remarked results of the research carried out on the subject are described, but this is not accompanied of specific citations that could facilitate the follow-up of these investigations. For instance, in point 4.5 regarding the implication of serotonergic transmission in the regulation of predawn TPR it would be appropriate to include citations of individual works that endorse the results. Something similar occurs in section 6, which refers to the involvement of the calcitonin receptor in the regulation of BTR in mice. At the end of paragraph 3 of this section it would be necessary to specify to which previous works the results cited in the text correspond.

We changed the citations on Page 8-10.

Figure 2 should be located later in the text, since part of its content, for example, the mechanisms that regulate the pre-dawn are exposed later in the text. A suitable place would be just before section 5. Mammalian BTR.

We assume that the reviewer indicates Fig. 7 not Fig.2, and therefore we placed Fig. 7 at just before section 5 on Page 9.